# Unique developmental trajectories of risk behaviors in adolescence and associated outcomes in young adulthood

**Margot Peeters**[1]*, **Albertine Oldehinkel**[2], **René Veenstra**[3], **Wilma Vollebergh**[1]

**1** Utrecht University, Utrecht, the Netherlands, **2** University Medical Center Groningen, Groningen, the Netherlands, **3** University of Groningen, Groningen, the Netherlands

* m.peeters1@uu.nl

## Abstract

This study aimed at assessing developmental trajectories of risk behaviors from adolescence into young adulthood and their associations with outcomes in young adulthood (i.e. education, employment). Data of the TRacking Adolescents' Individual Lives Survey (TRAILS) including 2,149 participants (mean age = 13.6, SD = 0.5, 51% girls) were used to examine the development of alcohol, cannabis, smoking, and externalizing behavior. The results showed that the associations between these risk behaviors varied with age, and revealed varying developmental patterns throughout adolescence. Most notably alcohol use did not covary strongly with the other risk behaviors. The often assumed peak in risk behavior in adolescence was only found in a small group, and only for alcohol (7.4%) and cannabis use (3.4%), but not for smoking or externalizing behavior. Most adolescents revealed only low involvement in risk behavior, with the largest differences between low and high trajectories emerging in late adolescence (> 19 years). Clustering of risk behavior throughout adolescence is rather the exception than the rule and depends on age and type of risk behavior. Differences in risk behavior between individuals become the largest in late adolescence, possibly influencing successful transition into adulthood visible in educational attainment and employment.

## Introduction

Risk behavior has been defined as reckless behavior [1] and as behavior that could lead to negative consequences [2]. Overall, the concept of risk behavior in adolescence has been used to refer to a collection of different behaviors, such as minor delinquency, aggression, risky sexual behavior, alcohol use, cannabis use, smoking, illicit drug use, and risky driving [3, 4, 5]. Several researchers have investigated the clustering of risk behaviors during adolescence [6–11]. Although these researchers have identified covariance between risk behaviors during certain periods of adolescence, involvement in risk behavior, and also the clustering of these behaviors, might still differ over the course of adolescence [12,13]. Some risk behaviors, such as aggression and minor delinquency, are more common in early adolescence whereas other risk

**Data Availability Statement:** Because TRAILS is a study that started 20 years ago (in 2001), the authors do not have permission to share the data because they did not ask participants to make their

data available. Given that participants have not given informed consent to have their personal data publicly shared, legal and ethical restrictions prevent the authors from making their data publicly available. Data are available upon request from the TRAILS data manager (trails@umcg.nl). Information about the consent method is available by the ethics committee; Central Committee on Research Involving Human subjects (CCMO; tc@ccmo.nl). For more details about the TRAILS study, please see https://www.toetsingonline.nl/to/ccmo_search.nsf/Searchform?OpenForm.

**Funding:** This research is part of the Consortium on Individual Development (CID). CID is funded through the Gravitation program of the Dutch Ministry of Education, Culture, and Science and the Dutch Organization of Scientific Research (NWO grant number 024.001.003). This research is further part of the TRacking Adolescents' Individual Lives Survey (TRAILS). Participating centers of TRAILS include various departments of the University Medical Center and University of Groningen, the University of Utrecht, the Radboud Medical Center Nijmegen, and the Parnassia Bavo group, all in the Netherlands. TRAILS has been financially supported by various grants from the Netherlands Organization for Scientific Research (NWO), ZonMW, GB-MaGW, the Dutch Ministry of Justice, the European Science Foundation, BBMRI-NL, and the participating universities. The funding organization had no role in the data collection, study design, manuscript preparation or in the decision to publish.

**Competing interests:** The authors have declared that no competing interests exist.

behaviors, such as alcohol or cannabis use, are more typical for late adolescence [14,15]. This implies that the involvement in risk behavior may not be captured in a stable and consistent construct over the course of adolescence and young adulthood. Engagement in risk behavior could vary during adolescents' development and the assumed underlying latent construct of risk behavior could vary accordingly. For instance, experimenting with alcohol use at 14 years of age could be risk behavior, whereas moderate alcohol use at age 19 might be relatively normative.

Some researchers have raised this issue of age-dependent involvement in risk behaviors [12,14,16], and revealed that the underlying construct indeed varied with age [9]. In line with research on the developmental stability of antisocial behavior [17], we examined whether the different risk behaviors can be grouped together as one underlying construct [8,18] or whether it would have been more adequate to examine specific risk taking behaviors separately [19,16].

Another question concerned the association between these risk behaviors–whether studied as one underlying construct or as specific risk behaviors–with outcomes in young adulthood (such as completing education), which is still inconclusive [20,21,22]. Possibly many adolescents engage in risk behavior only temporarily [23] for instance as a consequence of the changing social context and social role transitions (e.g., peers, work, high school, romantic relationships;[24,25,14,15]). Such temporary prevalence of risk behaviors might not necessarily be associated with young adult outcomes. Many studies [7,11,26] however covered relatively modest periods of time, only 2–4 years of development, which does not enable to capture temporary changes in the clustering of risk behavior during adolescence and young adulthood. Likewise, the associations with young adult outcomes cannot be examined with such studies.

To fill this gap, we investigated the development of four different risk behaviors, namely alcohol use, cannabis use, smoking behavior, aggression and minor delinquency, from early adolescence (around 14 years) to young adulthood (around 22 years). We examined whether these risk behaviors underlie the same latent construct and whether this construct was invariant (similar) over time and for girls and boys. This approach is similar to Odgers et al. (2008) who studied developmental trajectories of antisocial behavior in adolescence and related outcomes in young adulthood (e.g. education, employment).

## The conceptualization of risk behavior

Risk behavior has been studied in varies fields of research (e.g., epidemiology: [27,28]; developmental psychology: [29,13]; adolescent health:[30]; neuroscience: [31,32,10]; sociology:[33]). This resulted in diverging theoretical perspectives with respect to its conceptualization and operationalization [2,12]. A dominant theory of risk behavior in adolescence emerged from the neurocognitive field, suggesting that risk behavior in adolescence is the result of an imbalance between the development of behavioral control and the development of affective processes, such as reward and sensation seeking [10,31,34]. In neuroscience, the concept of risk behavior is often used to refer to risky decision-making processes; risky decision-making is seen as a proxy measure of real-life risk behavior [3]. The imbalance theory attempts to explain the increase in risk taking behavior in adolescence without differentiating between different kinds of risk behaviors. In contrast, adolescent health and epidemiological research have predominantly described and explained individual differences in the course and prevalence of risk behavior in terms of personality predispositions and differences in the school and family environment [13,30,35,36]. In the public health field, the concept of risk behavior has been used to refer to multiple health risk behaviors, such as substance use, aggression, sexual behavior, and unhealthy eating all in the naturalistic setting [18,31,37]. This overview illustrates that the conceptualization of risky behavior in adolescence depends on the field of interest. This

variation in conceptualization might relate to the different perspectives about the development of risk behavior and its consequences for adolescent health [16,36]. In this study we conceptualize risk behavior as behaviors that can be perceived as reckless and can have negative consequences for adolescent health. In this study we conceptualize risk behavior as behaviors that can be perceived as reckless and can have negative consequences for adolescent health [6–8].

For the investigation of the development of risk behavior over time it is important that the underlying construct of risk behavior is reflecting the same behavior over the course of adolescence and young adulthood [17]. Moreover, it is important to have an understanding whether all or only some risk behaviors contribute to certain outcomes in young adulthood. It is possible that some risky behaviors have a stronger negative impact on successful transition into young adulthood than other risk behaviors. There are some reasons to assume that the underlying construct of risk behavior would not be stable throughout adolescence and that possible related outcomes in young adulthood would differ as a function of the type of risk behavior as well as on the level of engagement. First, risk behaviors have unique characteristics contributing to varying (behavioral) consequences after engagement in these risk behaviors. Some risk behaviors are psychically addictive (e.g., smoking, illicit drug use) whereas others are not (e.g., aggression, minor delinquency). Some risk behaviors have immediate serious negative health effects (e.g., risky sexual behavior; risky driving; [36]), whereas others have delayed negative health effects (e.g., cannabis use, alcohol use). Some behaviors are normative and part of culturally appropriate behavioral patterns (e.g., having a drink at a party) and as such, they are not necessarily an expression of an underlying tendency to take risk [3]. In addition, international differences in alcohol and drug policy have a strong influence on what is perceived as norm-violating behavior and this policy perspective varies between risk behaviors as well between countries. For instance, in the United States, purchasing alcohol is legal at 21 years of age. In the Netherlands at 18 years (at the time we collected data in the cohort used in this study the legal drinking age was 16 years).

With respect to outcomes in young adulthood, considering these individual characteristics and changes in engagement in risk behavior might be important. Temporarily hazardous trajectories of risk behavior may, depending on the type of risk behavior, sometimes be normative, and associated negative consequences might not always be long-lasting [38,39]. Experimenting with alcohol or cannabis use might be relatively harmless [39] and even (socially) adaptive when it happens in a controlled manner and temporary. Some young adults outgrow these risk behaviors as soon as important role transitions that characterize young adulthood, such as completing a study or starting a job, become important in life [40].

Empirical studies investigating the relation between adolescents' risk behavior and outcomes in young adulthood are inconclusive and differ between types of risk behavior. Alcohol use for instance, has been identified as a risk factor [21] as well as a consequence of poor academic performance in mid-adolescence (15–17 years) [26]. Less positive educational outcomes were found for trajectories of binge drinking that were identified as heavy (i.e. increasing and late onset) in young adulthood (i.e. 21 years). In contrast, the early binge trajectory (decreasing again in late adolescence) did not reveal such relation with poorer educational outcomes in young adulthood [20]. Another study found that educational success depended on the type of risk behavior [22]. Negative impact on educational attainment was found for smoking and drug use, whereas binge drinking predicted lower school drop-out among high school and college students (18–25 years). In line with the latter study, educational attainment in young adulthood (around 25 years) was more weakly associated with the frequency of alcohol use before the age of 17 than of cannabis use in three different Australian cohort studies [41]. In sum, some typical involvement patterns in "risk behavior" could be perceived as risky. For some involvement patterns prolonged negative consequences interfering with a healthy

transition into adulthood may be absent whereas other patterns may have long-lasting negative effects on outcomes in young adulthood. It is conceivable that the long-term effects of adolescent risk behavior vary between, and depend on the level of engagement in risk behaviors.

### Present study

We examined the developmental trajectories of five kinds of risk behavior (i.e., aggression, minor delinquency, cannabis use, smoking and alcohol use) in the course of adolescence and young adulthood. We further investigated the associations between trajectories of risk behaviors from early adolescence (14 years) to late adolescence (22 years) and job/educational outcomes in young adulthood (26 years). The aim of this study was to:

1. determine whether there is one single time and sex invariant latent construct of risk behavior from early adolescence (14 years) to young adulthood (22 years);

2. model the developmental trajectory of risk behaviors throughout adolescence;

3. observe whether these trajectories predict outcomes (education, employment) in young adulthood (26 years).

## Method

### Participants

This study was a part of a national longitudinal cohort study, TRacking Adolescents' Individual Lives Survey (TRAILS). This longitudinal population study started in 2001/02 and included 2230 Dutch adolescents (born between October 1989 and September 1991) enrolled in study at age 11 (baseline). The assessment of these young adults (and their children) is still ongoing; at the most recent assessment wave (wave six) they were about 26 years old. The TRAILS study was conducted in accordance with the general ethical standards and was approved by the Central Committee on Research Involving Human subjects (CCMO). Children could participate after both their parents and they themselves provided consent and schools agreed to participate. In this particular study, waves 2 through 6 were included, because substance use questions in the first wave were brief because of the relatively young age at the first assessment (11 years). Each assessment took place approximately 3 years after the previous wave.

In total, 2,230 preadolescents were enrolled in the first wave, resulting in a sample with a mean age of 11.1 ($SD$ = 0.6) and comprising 51% girls. Wave 2 included 2,149 participants (96%) (mean age = 13.6, $SD$ = 0.5, 51% girls), wave 3 included 1,816 participants (81%; mean age = 16.3, $SD$ = 0.7, 52% girls), wave 4 included 1,881 participants (84%; mean age = 19.1, $SD$ = 0.6, 52% girls), wave 5 included 1,778 (80%) participants (mean age = 22.3, $SD$ = 0.6, 53% girls), and wave 6 included 1,618 (73%) participants (mean age = 25.7, $SD$ = 0.6, 55% girls). For a more detailed description of the cohort sample, selection criteria, and procedure, we refer to Oldehinkel and colleagues [42].

Attrition analyses comparing adolescents who participated in wave 6 with adolescents who dropped out in wave 6 or earlier, on risk behaviors (wave 2 to 5), sex, age, parental education, and single parenthood (wave 1) revealed several significant differences. Drop-outs were more likely to be male ($\chi^2$ (1, 2229) = 34.12, p < .01), were slightly older at wave 1 ($t$ (2227) = 2.759, cohen's d = .13), and were more likely to come from households in which parents were less educated ($t$ (2185) = -13,45, cohen's d = .64). In addition, the participants who dropped out smoked more across waves (wave 2: $t$ (1751) = 2.815, cohen's d: .17; wave 3: $t$ (1372) = 4.219, cohen's d: .34; wave 4: $t$ (1578) = 4.940, .43; wave 5: $t$ (1343) = 2.935, .38), used more alcohol in

wave 2 and wave 3 ($t$ (2058) = 2.156, cohen's d = .11 and $t$ (1623) = 3.810, cohen's d = .29), and exhibited more externalizing behavior in wave 3 ($t$ (1659) = 4.026., cohen's d = .28) and wave 4 ($t$ (1696) = 2.298, cohen's d = .19).

## Measures

**Risk behavior from 14 to 22 years.** **Alcohol use**. Participants indicated on how many days during the week (Monday to Thursday) and weekend (Friday to Sunday) they consumed alcohol on average. In addition, participants were asked to indicate the average number of drinks they consumed on a regular weekend or weekday (two items). We multiplied the drinking weekdays by the number of drinks consumed on a weekday and the drinking weekend days by the number of drinks on a regular weekend day (referring to a quantify-by-frequency measure). We specified a sum score by adding these two numbers together. Sum score reflect an average number of the consumed alcohol beverages during a regular week.

**Cannabis use**. Cannabis use was assessed by asking the participants to indicate the number of occasions (e.g., party, at home, going out) on which they consumed cannabis in the last month. Responses ranged from zero to forty times or more (0 to 10; 11–19; 205 20–39; 40 *or more*).

**Smoking**. Adolescents were asked to indicate the amount of cigarettes they smoked per day in the last 4 weeks. Response categories ranged from "never smoked" to "more than 20 cigarettes a day", with the two middle response categories distinguishing between occasional (e.g., once a week/one per day) and daily smokers (e.g., 2 to 20 cigarettes per day).

**Aggression and minor delinquency**. The Youth Self Report (YSR) and Adult Self Report (ASR, from 19 years onwards) were used to assess aggression and minor delinquency [43]. The scale included 29 items. Response categories for both subscales were, not true, somewhat true, and true, and respondents were asked to report their behavior in the past 6 months. A sample item of the aggression scale is "I am quick-tempered." A sample item for the minor delinquency scale is "I steal". Mean scores on both scales together were used as a measure of externalizing behavior. Both subscales revealed a good Cronbach's Alpha over all four waves, ranging from .80 to .85 for aggression and ranging from .70 to .77 for minor delinquency. Both subscales together form the externalizing behavior problems scale. We excluded three items on alcohol and drug use (compare to Monshouwer and colleagues [44] to avoid multi-collinearity between risk behaviors.

**Outcomes at 26 years.** The transition into young adulthood is often characterized by changes in relationships and work [45]. Since participants in our study were relatively young for marriage (the mean age in the Netherlands is 31 years for females and 34 for males [46], we only focused on education and employment at age 26.

**Study and educational level**. We determined educational level by the two questions assessing their current enrollment status and grade level as well as their highest degree obtained thus far. Missing information at wave 6 was supplemented with information from previous waves (e.g., highest educational degree), where possible. We created a dichotomous measure for both outcomes, indicating whether an adolescent was still studying (yes or no) and specifying the highest degree obtained (high; college or university degree, or low; secondary and vocational track).

**Unemployment**. For those who were not studying anymore, we determined whether they had a paid job. Adolescents indicated whether they had a paid job in the last month (yes or no).

**Confounders.** Demographic information about parents and family characteristics were obtained by self-report of the parents in the first wave. Parents reported the highest

educational level they completed (ranging from elementary school to university). Single parenthood was identified by the number of parents present in one household.

## Analyzing strategy

The analyses were divided into three parts:

1. A confirmatory factor analysis (CFA) was conducted to investigate the existence of an underlying latent factor of risk behavior. Alcohol use, cannabis use, smoking, aggression, and minor delinquency were included as latent indicators. A prerequisite for a general latent factor of risk behavior, is a stable invariant latent factor over the five waves which allows to compare latent factor scores between groups or over time. In other words, we need to ensure that we are not comparing apples with oranges [47]. To determine whether the latent construct of risk behavior was measurement invariant (MI) over time and invariant across sex, we constrained factor loadings (partial MI) and variances (full MI) for the four waves. In addition, we constrained the factors loadings for each sex and compared this model with a model without constraints. See S1 Fig for an overview of all steps.

2. A latent growth mixed model was used to evaluate latent classes of growth trajectories over time. Intercepts were freely estimated between classes, and slope variance were held equal (model fit dropped and convergence issues emerged when freeing the variances as well between classes). The optimal amount of classes was determined by (a) an increase of model fit indicated by the Akaike Information index (AIC) and the Bayesian Information Index (BIC); (b) an acceptable level of classification indicated by the entropy value ($>.80$); (c) a significant increase of fit indicated by the Bootstrap Likelihood Ratio Test (BLRT); (d) an acceptable sample size for each class ($> 2\%$; see also [48,49]). If the entropy is high enough (i.e., $>.80$), transporting patterns to other statistical programs is allowed [50].

3. Outcomes at age 26 were evaluated in relation to the risk behavior trajectories from 14 to 22 years. Multiple logistic regression was used to determine the chance that someone in a certain trajectory would score higher or lower on important outcomes in adulthood, such as study, work, and educational level. Environmental predispositions, such as lower SES and single parenthood in the family of origin, could affect adolescents' engagement in risk behavior as well as health outcomes in young adulthood [51]. Therefore, repeated analyses included confounders, such as adolescents' age, sex, parental education, and single parenthood in the family of origin. We corrected for multiple testing using the Bonferroni method.

Steps 1 and 2 were performed with Mplus version 8.0 using full information maximum likelihood (FIML) to deal with missing data for the risk behavior trajectories. Maximum likelihood with robust standard errors (MLR) was used as estimation method. For the third step, we saved the class membership with the highest probability and imported it to SPSS (compare Peeters et al., 2014[49]) to perform logistic regression analyses. For the risk behavior trajectories, no data was missing, as FIML was available in Mplus to handle the missing data. To avoid that trajectories of risk behavior were predicted while accounting for the outcomes at 26 or covariates specified in the model–this will happen when variables are added to the growth model—information on most likely trajectory membership for each participants was transported to SPSS. For outcomes at 26 years, approximately 40% of the data collected using the self-reported measures was missing. Attrition analyses suggested that adolescents who dropped-out of the study were more likely to be engaged in some risk behaviors at previous waves (2 to 5). Because no information about unemployment and education was available for

this particular group, it was not possible to compare this group with adolescents who still participated in the TRAILS study on the outcome variables. Hence, particular adolescents in the higher risk behavior trajectories might not have been included in the analyses focusing on the outcomes at 26.

## Results

### CFA results

The CFA analyses revealed four findings (S1 Table):

1. Factor loadings for aggression and minor delinquency were high (.70-.80), suggesting a strong overlap between the two indicators of risk behavior. We therefore used the combined factor of externalizing behavior in further analyses (as described by the ASEBA manual [43]).

2. The model fit reached an acceptable level of fit only when alcohol was removed as indicator (CFI = .92, RMSEA = .047). Factor loadings for alcohol use dropped below acceptable levels (.18, .17 and .12 for age 16, 19 and 22), when factor loadings were constrained over time (see S2 Table) and model fit dropped below acceptable levels when both intercepts and factor loadings were constrained to be equal across waves (see S3 Table).

3. Continuing with a model without alcohol use, we did not find evidence for measurement invariance only for partial measurement invariance (factor loadings constrained, but intercepts not, CFI = .957, RMSEA = .037). By violating the assumption of scalar variance (referring to intercepts constrained), it is possible that the relative value on the latent construct differs from the item indicators underlying this construct. This often indicates developmental variation (for example, smoking could have a high value at wave 2, but moderate value at wave 4 but still contribute in a similar matter to the latent construct; [52]). As a result, trajectories of the latent construct of risk behavior will not represent clustering of risk behavior (e.g., all high), but rather typical patterns in behavior that tend to co-occur more often during a certain period in adolescence. Although for some studies this might not be a problem [47], for our study it will not shed light on the question whether risk behaviors clusters together in a similar way from early to late adolescence.

4. Although model fit measures slightly favored an unconstrained model (sex differences; CFI = .881 vs .879, RMSEA = .058 vs no sex differences; RMSEA = .058), factor loadings for the female group were non-significant for cannabis use on all waves and for externalizing problems on the first wave. Therefore, we assumed no sex differences in the construct of risk behavior (see S4 Table).

We were unable to fit a model, in which a meaningful and stable latent construct of risk behavior could be defined. Further analyses included the four risk behaviors (alcohol, cannabis, smoking, and externalizing behavior) separately. Because smoking as well as cannabis use included many zero counts and overdispersed data, we used a negative binominal model for these two risk behaviors [53]. For alcohol use, a count model (Poisson distribution) was used, as the number of zero counts was not reaching similarly high levels as for smoking and cannabis.

### Results growth trajectories

**Descriptive statistics trajectories.** Tables 1 through 4 depict the descriptive statistics of the trajectories. For alcohol use, we found four trajectories (Fig 1): stable low trajectory (38%),

**Table 1. Descriptive statistics for alcohol per trajectory.**

|  | Stable Low (N = 778; 38%) Mean (SD) | Moderate increasing (N = 806; 39%) Means (SD) | Heavy increasing (N = 325; 16%) Mean (SD) | Peaking (N = 148; 7%) Mean (SD) |
|---|---|---|---|---|
| Alcohol 14 years | 0.23 (0.66) | 0.90 (1.47) | 2.01(2.75) | 10.21 (8.26) |
| Alcohol 16 years | 1.02 (1.55) | 4.62 (3.47) | 11.27 (7.24) | 17.21 (9.49) |
| Alcohol 19 years | 2.00 (1.98) | 7.31 (4.21) | 18.26 (9.02) | 10.23 (6.45) |
| Alcohol 22 years | 2.29 (2.10) | 8.38 (4.64) | 20.80 (8.95) | 6.96 (4.63) |

**Table 2. Descriptive statistics for cannabis per trajectory.**

|  | Never use (N = 1654; 76%) Mean (SD) | Low (N = 221; 13%) Mean (SD) | Late increase (N = 62; 4%) Mean (SD) | Peaking (N = 55; 3%) Mean (SD) | Early Increase (N = 57; 4%) Mean (SD) |
|---|---|---|---|---|---|
| Cannabis 14 years | 0.01 (0.14) | 0.48 (1.83) | 0.25 (0.60) | 0.66 (5.08) | 1.51 (5.12) |
| Cannabis 16 years | 0.05 (0.24) | 2.23 (3.75) | 2.12 (2.82) | 14.34 (16.48) | 12.91 (15.67) |
| Cannabis 19 years | 0.05 (0.23) | 2.39 (2.77) | 8.80 (7.83) | 13.11 (12.86) | 28.51 (15.05) |
| Cannabis 22 years | 0.04 (0.21) | 2.30 (2.59) | 21.31 (15.18) | 7.68 (4.98) | 25.68 (16.42) |

moderate increasing trajectory (39%), peaking trajectory (7%), and a heavy increasing trajectory (16%). All trajectories described an increase in alcohol use during the adolescence, with the exception of the peaking trajectory, which revealed a decline in late adolescence and young adulthood. We found five different trajectories for cannabis (Fig 2): a never use (76%), low (13%), peaking (3%), early increase (4%) and late increase (4%). The never use and low cannabis trajectory revealed a (small) increase until late adolescence (e.g., around 19 years) followed by decline (see Table 2). The late increasing trajectory continued to rise until the age of 22.

**Table 3. Descriptive statistics for smoking behavior by each trajectory separately.**

|  | Stable Low (N = 1299; 61%) Mean (SD) | Moderate increasing (N = 400; 22%) Mean (SD) | Heavy increasing (N = 311; 17%) Mean (SD) |
|---|---|---|---|
| Smoking 14 years | 0.01 (0.12) | 1.08 (2.41) | 5.19 (8.72) |
| Smoking 16 years | 0.05 (0.22) | 4.18 (3.95) | 13.07 (8.87) |
| Smoking 19 years | 0.10 (0.32) | 6.27(4.76) | 15.50 (7.64) |
| Smoking 22 years | 0.25 (0.68) | 7.83 (5.09) | 16.65 (7.05) |

**Table 4. Descriptive statistics for externalizing behavior by each trajectory separately.**

|  | Low (N = 1854; 87%) Mean (SD) | High (N = 211; 13%) Mean (SD) |
|---|---|---|
| Externalizing behavior 14 years | 0.29 (.20) | 0.46 (.24) |
| Externalizing behavior 16 years | 0.29 (.19) | 0.56 (.23) |
| Externalizing behavior 19 years | 0.19 (.18) | 0.55 (.26) |
| Externalizing behavior 22 years | 0.15 (.12) | 0.59 (.16) |

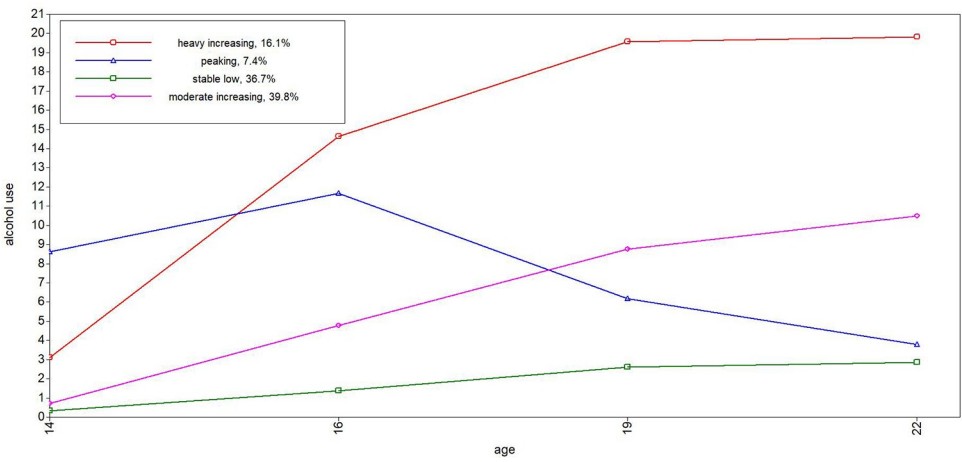

**Fig 1. Trajectories of alcohol use from 14 to 22 years.**

Three smoking trajectories were found (Fig 3): stable low (61%), moderate increasing (22%), and heavy increasing (17%). All smoking trajectories increased until 22 years of age, although lower trajectories (low and moderate) exhibited a much smaller increase compared to the heavy smoking trajectory. We found two trajectories for externalizing behavior (Fig 4): low (87%) and high (13%). The low trajectory for externalizing behavior decreased after mid adolescence, with continuing lower levels of externalizing behavior in late adolescence (19 years) and young adulthood (22 years), whereas the high trajectory revealed a mild increase during adolescence until 22 years.

In sum, little to almost no involvement in risk behavior was found for the largest group of adolescents. In general, risk behavior increased steadily in early and mid-adolescence, leading to more pronounced differences between risk behavior trajectories in late adolescence and young adulthood than before. Diverging trajectories with increasing age were also observed for externalizing behavior. Lastly, only a minority of the adolescents revealed a peak in risk taking behavior for alcohol (7,4%) and cannabis use (3.4%).

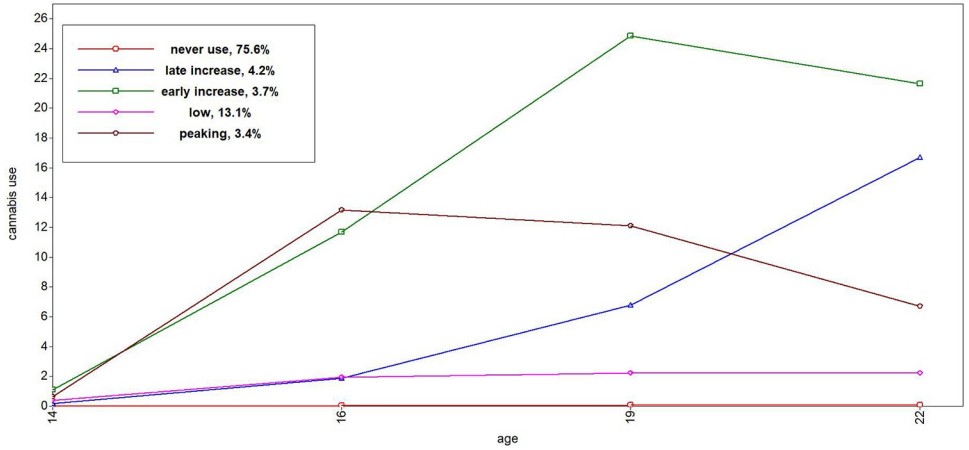

**Fig 2. Trajectories of cannabis use from 14 to 22 years.**

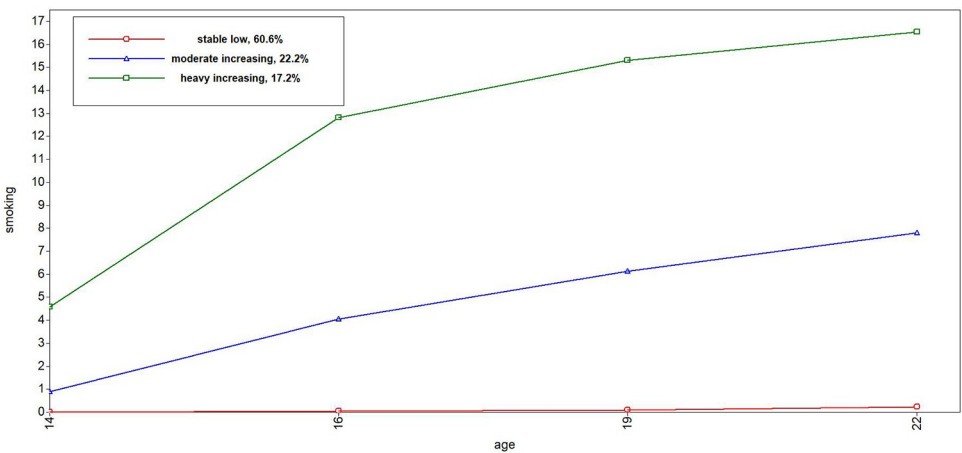

**Fig 3. Trajectories of smoking behavior from 14 to 22 years.**

## Outcomes at age 26

We evaluated trajectories of risk behavior in relation to study completion, having a job, and highest educational level completed. We performed all analyses with and without confounders (age, sex, educational level of parents, and single parenthood of the family of origin). Logistic regression results for each separate trajectory in relation to outcomes in young adulthood are presented in Tables 5 through 8.

With respect to alcohol use (Table 5), significant differences between trajectories emerged only for educational level, with the peaking trajectory being more likely to include less educated participants than the stable low trajectory. For cannabis use (Table 6), adolescents in the three highest trajectories (referring to the increasing, peaking, and early onset trajectories) were significantly more likely to be low educated and less likely to have a job than the stable low trajectory. For smoking behavior (Table 7), adolescents in the heavy smoking trajectory were less likely to study or have a job and were less educated compared to adolescents in the low stable trajectory. Adolescents in the moderate increasing smoking trajectory were less educated and less likely to have a job than the stable low trajectory group of smoking.

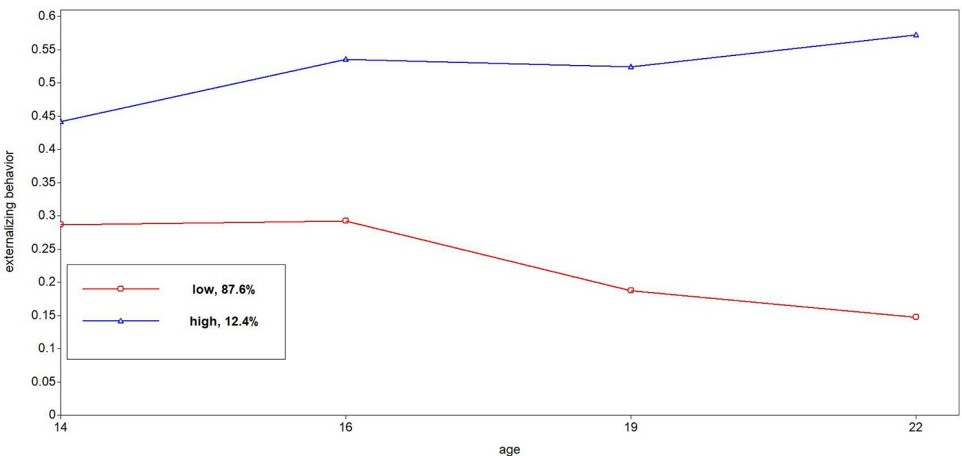

**Fig 4. Trajectories of externalizing behavior from 14 to 22 years.**

**Table 5. Logistic regression with and without confounders for trajectories of alcohol use.**

| Trajectory Outcome at 26 | | Trajectory class alcohol N (percentage) | | | | Comparison with "norm class = low" Without confounders Odds (CI) | | | Comparison with "norm class = low" With confounders Odds (CI) | | |
|---|---|---|---|---|---|---|---|---|---|---|---|
| | | Low | Moderate | Increase | Peak | Low vs moderate | Low vs increase | Low vs Peak | Low vs moderate | Low vs increase | Low vs Peak |
| Study | | | | | | 1.10 (0.86–1.40) | 1.25 (0.88–1.76) | 0.62 (0.36–1.09). | 0.99 (0.76–1.29) | 0.94 (0.63–1.40) | 0.64 (0.36–1.15) |
| | No | 272 (67%) | 411 (44%) | 130 (62%) | 61 (76%) | | | | | | |
| Education | | | | | | 0.91 (0.72–1.14) | 0.88 (0.63–1.22) | 0.46* (0.27–0.77) | 0.82 (0.63–1.06). | 0.71 (0.47–1.06) | 0.50 (0.29–0.88) |
| | Lower | 211 (52%) | 333 (52%) | 116 (55%) | 56 (70%) | | | | | | |
| No job[a] | | | | | | 0.75 (0.52–1.10) | 1.19 (0.71–1.99) | 0.56 (0.24–1.31) | 0.82 (0.54–1.24) | 1.49 (0.80–2.79) | 0.50 (0.20–1.28) |
| | Yes | 51 (19%) | 54 (13%) | 28 (21%) | 7 (11%) | | | | | | |

[a] only adolescents included who indicated that they were not studying anymore;

Bonferroni correction for multiple testing; p < .016*

For externalizing behavioral problems (Table 8), the low stable trajectory significantly differed from the higher trajectory on all three outcomes. Adolescents in the high trajectory of externalizing problems were more likely to study at age 26, but were overall less educated, and were less likely to have a job. Repeated analyses with confounders revealed similar results.

## Discussion

Following up a large cohort of adolescents into young adulthood, our study revealed that the associations between specific risk behaviors tend to vary with age; we did not find a single

**Table 6. Logistic regression with and without confounders for trajectories for cannabis use.**

| Trajectory Outcome at 26 | | Trajectory class cannabis N (percentage) | | | | | Comparison with "norm class = low" Without confounders Odds (CI) | | | | Comparison with "norm class = low" With confounders Odds (CI) | | | |
|---|---|---|---|---|---|---|---|---|---|---|---|---|---|---|
| | | Never | Low | Late increase | Peak | Early increase | Never vs low | Never vs late increase | Never vs Peak | Never vs early increase | Never vs low | Never vs late increase | Never vs Peak | Never vs early increase |
| Study | | | | | | | 1.32 (0.93–1.86) | 0.83 (0.38–1.83) | 1.07 (0.52–2.19). | 0.90 (0.45–1.80) | 1.31 (0.91–1.87) | 0.71 (0.31–1.66) | 1.08 (0.51–2.28) | .89 (0.43–1.82) |
| | No | 718 (66%) | 91 (59%) | 21 (70%) | 21 (64%) | 25 (68%) | | | | | | | | |
| Education | | | | | | | 0.73 (0.52–1.03) | 0.26* (0.10–0.63) | 0.40* (0.18–0.86) | 0.21* (0.09–.50) | 0.70 (0.48–1.02) | 0.26* (0.10–0.64) | 0.37 (0.17–0.89) | 0.21* (0.08–0.52) |
| | Lower | 550 (51%) | 89 (58%) | 24 (80%) | 24 (73%) | 31 (84%) | | | | | | | | |
| No job[a] | | | | | | | 1.09 (.59–1.99) | 2.99 (1.18–7.57) | 3.64* (1.48–8.97) | 3.94* (1.73–8.99) | 1.17 (0.62–2.20) | 3.41 (1.32–8.83) | 3.83* (1.46–10.06) | 4.23* (1.74–10.28) |
| | Yes | 103 (14%) | 14 (15%) | 7 (33%) | 8 (38%) | 10 (40%) | | | | | | | | |

[a] only adolescents included who indicated that they were not studying anymore;

Bonferroni correction for multiple testing; p < .016*

**Table 7. Logistic regression with and without confounders for trajectories of smoking.**

| Trajectory Outcome at 26 | | Trajectory class smoking N (percentage) | | | Comparison with "norm class = low" Without confounders Odds (CI) | | Comparison with "norm class = low" With confounders Odds (CI) | |
|---|---|---|---|---|---|---|---|---|
| | | Low | Moderate | Heavy | Low vs moderate | Low vs heavy | Low vs moderate | Low vs heavy |
| Study | | | | | 0.83 (0.62–1.12) | 0.58* (.039–0.82) | 0.87 (0.64–1.17) | 0.65 (0.44–0.98) |
| | Yes | 331 (37%) | 86 (33%) | 42 (25%) | | | | |
| Education | | | | | 0.29* (0.21–0.39) | 0.14* (0.09–0.21) | 0.28* (0.20–0.38) | .16* (0.10–0.25) |
| | Lower | 379 (42%) | 189 (72%) | 143 (84%) | | | | |
| No job[a] | | | | | 1.88* (1.21–2.92) | 2.57* (1.62–4.10) | 1.93* (1.22–3.05) | 2.44* (1.47–4.08) |
| | Yes | 70 (12%) | 37 (21%) | 34 (27%) | | | | |

[a] only adolescents included who indicated that they were not studying anymore;

Bonferroni correction for multiple testing; p < .016*

underlying risk behavior construct throughout adolescence. Therefore, we examined trajectories for specific risk behaviors. In contrast to what is often assumed, the 'peak' in risk behaviors in mid and late adolescence was not common [10,31,34]; it was, only found for a very small minority of the adolescents, and only for alcohol and cannabis use. In contrast, a continuing increase after mid adolescence was found for much larger groups of adolescents, and the majority of adolescents fell into consistently abstaining or low trajectories. As a result, the difference in the prevalence of the specific risk behaviors between adolescents in the various trajectories was persistently and substantially larger in early adulthood than in early adolescence.

With respect to the first conclusion, combining all risk behaviors (alcohol, cannabis, smoking, externalizing behavior) in one model did not produce good model fit (poor model fit indexes and low factor loadings), indicating that a single construct does not account for the individual differences observed in the four risk behaviors from early to late adolescence. Clustering of risk behaviors during adolescence might be observed during some phases of adolescence, as former research using much shorter periods has convincingly shown [6,7,44]. However, the developmental differences and diversity in trajectories of risk behavior indicate that the underlying construct of risk behavior is not the same throughout adolescence and young adulthood.

**Table 8. Logistic regression with and without confounders for trajectories of externalizing behavior.**

| Trajectory Outcome at 26 | | Trajectory class externalizing behavior N (percentage) | | Comparison with "norm class = low" Without confounders Odds (CI) | Comparison with "norm class = low" With confounders Odds (CI) |
|---|---|---|---|---|---|
| | | Low | high | Low vs high | Low vs high |
| Study | | | | 1.54 (1.08–2.20) | 1.67* (1.14–2.43) |
| | Yes | 405 (34%) | 61 (44%) | | |
| Education | | | | 0.3* (0.23–0.50) | 0.34* (0.22–0.53) |
| | Lower | 613 (51%) | 105 (75%) | | |
| No job[a] | | | | 3.14* (1.90–5.22) | 3.06* (1.77–5.31) |
| | Yes | 115 (14%) | 27 (35%) | | |

[a] only adolescents included who indicated that they were not studying anymore;

Bonferroni correction for multiple testing; p < .016*

Alcohol use fitted poorly in the assumed latent construct of risk behavior. Whereas factor loadings for alcohol use were only acceptable at age 14 when part of a latent construct of risk behavior; factor loadings as well as model fit dropped below acceptable levels after age 14. This finding suggests that alcohol use could be seen as a risk behavior in early adolescence, but is becoming rather normative at age 16. Note that our study was conducted in the Netherlands, a country that had a history of being lenient with respect to adolescent drinking (ESPAD group [54]). In addition, our study participants were adolescents in the first decade of this century. The remarkable decrease in alcohol consumption found in various countries in Europe, most notably in the Netherlands, was in later years [55]. As a result, for our cohort alcohol use was already normative behavior at young ages. This implicates that it is not alcohol use as such, that should be considered as marker of risk behavior in adolescents, but only alcohol use in a context in which it is non-normative and in which it is not allowed for adolescents to drink [9]. To ascertain that our measure of alcohol use reflected the entire spectrum of drinking behavior (e.g., weekly and heavy episodic drinking), we repeated our analyses, including drunkenness as a latent factor, which revealed similar results (data can be requested from the first author).

In sum, the findings of this study reveal that the observed risk behaviors throughout adolescence do not tap consistently in the same underlying construct of risk behavior. There might be clustering of risk behavior during some phases of adolescence, however, the absence of measurement invariance over time, also visible in the varying developmental patterns of the individual risk behaviors, indicate that co-occurrence of risk behaviors is not consistent throughout adolescence. In a similar study [56] it was found that symptoms of nicotine, alcohol and cannabis dependence and abuse clearly clustered together in adolescence (14–17 years), but not so much in young adulthood (22–29 years. It is recommended to take this finding into account when investigating risk behavior in laboratory settings, such as often done in the neurocognitive field of research [31,34] because the decision process to engage in risk behavior might vary in a similar way during adolescence and young adulthood [12].

In contrast to research that indicates a peak in risk behavior in mid- and late adolescence for most adolescents (16–20 years;[10,31,34]), we observed the peak in risk behavior only for alcohol and cannabis use and only for a small minority of adolescents (7.3% and 3.4% respectively). These findings are in line with several other trajectory papers on alcohol use that also found the assumed peak only for a small minority [13,57]. In our study, the minority of adolescents in the peaking trajectory had an early onset of alcohol and cannabis use, which peaked around 16–19 years of age and declined in late adolescence and young adulthood. The majority of the adolescents had patterns of risk behaviors that remained stable tended to increase until late adolescence and young adulthood (up till 22 years; [24,39,58]).

In general, our findings revealed a growing disparity in risk behaviors during adolescence. In other words, the development of risk behavior in adolescence and young adulthood seems to be characterized by "diverging pathways," with the difference between heavy, moderate, and low engagement in risk behavior becoming larger as adolescents grow older. Also externalizing problems showed a pattern with substantially diverging high and low trajectories in the course of ten years. That finding is in contrast with studies observing two additional trajectories (i.e. increasing and decreasing) of antisocial behaviors [17]). This contrasting finding could have been a result of the fact that we excluded substance use items form the externalizing subscale to avoid multicollinearity.

## Outcomes at age 26

Except for alcohol use, the trajectories reflecting the heaviest involvement in risk behavior predicted the least favorable outcomes (e.g., unemployment, lower education). These unfavorable

outcomes were probably not due to already existing environmental adversities, as lower parental education or single parent household around age 11 were both controlled for in the analyses. This suggests that a disadvantaged position in young adulthood could be a result of cumulative effects of risk behavior. Alcohol use trajectories did not differentiate between adolescents developing successfully into young adults and adolescents who experienced difficulties in transitioning into adult roles. Thus drinking alcohol apparently does not present a risk for the pertinent outcomes. This may be due to the fact that drinking alcohol has been quite normative for adolescents in the Netherlands, in particular in the TRAILS-cohort [54].

Remarkably, with respect to alcohol use, the "peak" trajectory was associated with lower educational achievement at age 26 (it should be noted that this effect disappeared after Bonferroni corrections and controlling for other covariates). Additional analyses revealed that at 19 years, adolescents in this particular trajectory work on average more hours than adolescents in the other trajectories (mean hours stable low = 15, moderate increasing = 16, heavy increasing = 19 and peaking = 22). The responsibilities that come with the labor market entry could be the reason for the decline in drinking behavior in this group [24,57,58]. This trajectory showed no increased risk of unemployment, further supporting this notion. In addition, adolescents in the heavy drinking trajectory were not lower educated nor were they at an increased risk of unemployed compared to the lower drinking trajectories. This finding is consistent with research revealing an increase in alcohol use in late adolescence [14,15] as well as research suggesting a relatively weak associations between educational level or socio-economic status and alcohol use [30,41,57]. The findings of our study suggest that drinking trajectories in adolescence reflect changing social and cultural contexts in which earlier transition to adult roles, such as work, could be typical for the lower socioeconomic strata rather than alcohol use per se [57]. However, further research on trajectories of alcohol use should look at influences of (changing) socioeconomic status and education throughout adolescence and into young adulthood to support this line of reasoning.

For cannabis, smoking and externalizing trajectories, heavy engagement was associated with an increased likelihood for lower education and unemployment. These results remained significant after controlling for confounding variables such as parental education and being raised in a single parent household. This suggests that heavy cannabis use, externalizing behavior and smoking are possible indicators for less successful adult role transitioning in young adulthood. Future research could include other markers of adulthood such as marriage, children and financial situation [23], to investigate whether the negative impact of heavy cannabis, smoking and externalizing behavior affects other aspects of adulthood as well.

## Limitations

The findings of this study should be interpreted in light of some limitations. First, we included no information about risk behavior after the age of 22, as we wanted the trajectories to precede the outcomes in young adulthood. Maturing out of alcohol use, for instance, may occur after the age of 22 [24,58]. Therefore, some adolescents in the heavy drinking trajectory might have decreased their use after the age of 22. We cannot rule out the possibility that continued engagement in risk behavior after the age of 22 could have generated different trajectories for which association with less favorable outcomes at 26 years would have been different. Nevertheless, for smoking, cannabis, and externalizing behavior, the picture that emerged was clear, with odds of less favorable outcomes increasing for the higher risk behavior trajectories. A second limitation of our study was that we did not analyze the pathways of education that may be associated with the educational outcomes at age 26. In the Netherlands, selection of adolescents into different educational tracks (differentiating four different tracks from vocational

training to pre-university education) takes place in early adolescence, at age 12. As a result, those who completed lower education at age 26, were most likely to be in lower educational tracks throughout adolescence. Thus, the association between risk behaviors and young adult outcomes reflects this association and should not be interpreted as causal. Reverse causality could be an explanation as well, also because engagement in certain risk behaviors could also be a result of difficulties with academic performance [26] or holding employment (e.g., self-medication, coping). For further research, we recommend to analyze the unique contribution of risk behavior trajectories to adverse outcomes in young adulthood when considering simultaneous developmental patterns in educational level in the course of adolescence. Third, future research could include a more ethnically diverse population (in our sample only 10% of the parents had a minority background) to investigate whether results are similar for other ethnic groups. Research shows that alcohol consumption for instance is less common among young adolescents with a minority background [59], possibly because of religious considerations. Generalizability problems may also arise for other adolescents in other countries, as drinking culture differ among countries and legal policies can have an impact on legalization of drinking at a certain age [54,55].

Lastly, data from all adolescents were included in the trajectory analyses, as we used FIML to handle the missing data; however, we excluded dropouts from the analysis at age 26. Attrition analysis revealed somewhat higher rates of risk behavior for the drop-outs, possibly indicating underestimation of the number of adolescents in the heaviest risk taking group as well as a bias in the observed association with less favorable outcomes at age 26. However, based on the missing data analyses, it is likely that the observed associations between high involvement in risk behavior and less favorable outcomes at age 26 would have been more strongly, had all adolescents remained in the analysis.

## Conclusion

Although the term risk-taking behaviors is often used to refer to a large variety of behaviors, hereby (implicitly) assuming that they reflect the same underlying tendency or behavioral syndrome, our findings provided neither evidence for such a tendency nor for a consistent clustering of risk behaviors throughout adolescence and young adulthood (compare; [6,7,8]). In particular alcohol use was not strongly associated with the other indicators of risk behavior. We did not find a clear peak in risk behaviors in middle adolescence, except for alcohol and cannabis use in a small minority of the participants. We found that the specific risk behaviors (e.g., alcohol, cannabis, smoking, and externalizing behaviors) follow unique developmental patterns with growing disparities between low and high levels of involvement, and only the highest involvement in risk behavior was associated with adverse outcomes in young adulthood, again except for alcohol use. Examining risk behavior as a single construct may not do justice to the different facets of risk behavior that might change in response to varying norms and changing social contexts typical for adolescent development.

These result suggest that focusing on alcohol use in adolescence as possible marker for negative outcomes in young adulthood will not be the best approach to identify adolescents at risk for later problems in young adulthood. By no means we want to imply that the chosen outcomes are exhaustive in predicting positive outcomes in young adulthood, though we believe work and education are important markers for successful transition into young adulthood [20,21,22]. For policy and intervention purposes, it may be more efficient to focus on other risky behaviors, such as cannabis use or externalizing problems. More particular, is may be wise to focus on the heavy, persistent trajectories of risky behaviors to identify the adolescents most at risk for being unsuccessful in their transition into young adulthood.

## Supporting information

**S1 Table. Confirmatory factor analysis (CFA) for each wave separately.**
(DOCX)

**S2 Table. Measurement invariance (MI): Partial MI; factor loadings constraint to be equal.** [a]: model results for 14 and 16 years [b]: model results for 16 and 19 years [c]: model results for wave 14 to 19 years [d]: model results for 14 to 22 years.
(DOCX)

**S3 Table. Measurement invariance: Full MI; factor loadings and intercepts constrained to be equal.** [a]: model results for 14 and 16 years [b]: model results for 16 and 19 years [c]: model results for wave 14 to 19 years [d]: model results for 14 to 22 years.
(DOCX)

**S4 Table. Sex differences with partial MI and without alcohol (only factor loadings constrained).**
(DOCX)

**S1 Fig. Path diagram about decisions CFA analyses.**
(DOCX)

## Acknowledgments

This research is part of the Consortium on Individual Development (CID). CID is funded through the Gravitation program of the Dutch Ministry of Education, Culture, and Science and the Dutch Organization of Scientific Research (NWO grant number 024.001.003). This research is further part of the TRacking Adolescents' Individual Lives Survey (TRAILS). Participating centers of TRAILS include various departments of the University Medical Center and University of Groningen, the University of Utrecht, the Radboud Medical Center Nijmegen, and the Parnassia Bavo group, all in the Netherlands. TRAILS has been financially supported by various grants from the Netherlands Organization for Scientific Research (NWO), ZonMW, GB-MaGW, the Dutch Ministry of Justice, the European Science Foundation, BBMRI-NL, and the participating universities. We are grateful to everyone who participated in this research or worked on this project to and make it possible.

## Author Contributions

**Conceptualization:** Margot Peeters, René Veenstra, Wilma Vollebergh.

**Formal analysis:** Margot Peeters, Wilma Vollebergh.

**Funding acquisition:** Albertine Oldehinkel, René Veenstra, Wilma Vollebergh.

**Methodology:** Margot Peeters, Wilma Vollebergh.

**Supervision:** Albertine Oldehinkel, Wilma Vollebergh.

**Writing – original draft:** Margot Peeters, Albertine Oldehinkel, René Veenstra, Wilma Vollebergh.

**Writing – review & editing:** Margot Peeters, Albertine Oldehinkel, René Veenstra, Wilma Vollebergh.

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
