## [Decision Letter · Decision Letter 0]

14 Aug 2019

PONE-D-19-17958

Unique Developmental Trajectories of Risk Behaviors in Adolescence and Associated Outcomes in Young Adulthood

PLOS ONE

Dear Dr. Peeters,

Thank you for submitting your manuscript to PLOS ONE. After careful consideration, we feel that it has merit but does not fully meet PLOS ONE’s publication criteria as it currently stands. Therefore, we invite you to submit a revised version of the manuscript that addresses the points raised during the review process.

We would appreciate receiving your revised manuscript by Sep 28 2019 11:59PM. To enhance the reproducibility of your results, we recommend that if applicable you deposit your laboratory protocols in protocols.io, where a protocol can be assigned its own identifier (DOI) such that it can be cited independently in the future. For instructions see: http://journals.plos.org/plosone/s/submission-guidelines#loc-laboratory-protocols

We look forward to receiving your revised manuscript.

Kind regards,

Geilson Lima Santana, M.D., Ph.D.

Academic Editor

PLOS ONE

Journal Requirements:

1. We note that you have indicated that data from this study are available upon request. PLOS only allows data to be available upon request if there are legal or ethical restrictions on sharing data publicly. For more information on unacceptable data access restrictions, please see http://journals.plos.org/plosone/s/data-availability#loc-unacceptable-data-access-restrictions.

Reviewers' comments:

Reviewer's Responses to Questions

**Comments to the Author**

1. Is the manuscript technically sound, and do the data support the conclusions?

Reviewer #1: Yes

Reviewer #2: Partly

2. Has the statistical analysis been performed appropriately and rigorously? 

Reviewer #1: Yes

Reviewer #2: Yes

3. Have the authors made all data underlying the findings in their manuscript fully available?

Reviewer #1: No

Reviewer #2: No

4. Is the manuscript presented in an intelligible fashion and written in standard English?

Reviewer #1: Yes

Reviewer #2: Yes

5. Review Comments to the Author

Reviewer #1: The paper is well written and method is sound. However, the description of statistical analysis and results presentation need minor revision before publication.

Reviewer #2: Review of PONE-D-19-17958 “Unique Developmental Trajectories of Risk Behaviors in Adolescence and Associated Outcomes in Young Adulthood.”

The purpose of this study was to identify unique developmental trajectories of risk behaviors and associations with outcomes using a large, longitudinal study of Dutch adolescents (TRAILS). Peak risk behavior occurred in late adolescence (>19 years), thus this is where the largest differences in risk behavior trajectories was demonstrated. The authors conclude this likely influences the successful transition into higher educational attainment and gainful employment. The authors also note that alcohol did not covary with other risk behaviors (cannabis, smoking, externalizing) and present this as a key finding. There are many strengths of the study design, measures, and analyses. I have a few concerns I outline below that I believe if addressed, would strengthen the manuscript substantially.

The intro lays out the rationale well but ignores research that has demonstrated the clustering of risk and externalizing behaviors, including different kinds of substance use, is stronger in adolescence and dissipates over time (people specialize in one drug, etc.). See Vrieze et al (Am J Psychiatry 2012; 169:1073–1081) for a discussion on this. I encourage the authors to consider integration of this perspective into your rationale and findings.

Method

The attrition analyses would be more helpful if the authors could comment on the nature of the effect size rather than focusing on p-values alone. How big of an effect might this have on generalizing results to the larger population?

Were the risk outcomes (e.g., cannabis) log-transformed to better approximate normality assumptions? Cannabis in particular seems quite skewed.

Was there any variability in race/ethnicity of the sample? This might be included as a covariate/confounder, if so.

Results

The authors should present more details about their CFA model where alcohol was dropped (what were the factor loadings for the other risk behaviors? What were the loadings for males vs. females?). Why not evaluate separate CFA models at the different ages of assessment to compare how these models fit at unique developmental periods? In reference to the Vrieze et al. article above, you may find good fit at some point in adolescence but not later in young adulthood.

For the group names for risk behavior, the ones labelled “moderate” seem to me would be better reflected with the name “moderate-increasing” (for alcohol, smoking) and the group labeled “moderate” for cannabis appears to just be “low” whereas “stable low” is essentially “never use.”

The figures demonstrating the trajectory groups are extremely hard to read (fuzzy).

Given the large number of tests (comparing each trajectory for each risk behavior group to one another in relation to the three education and employment outcomes), do the authors worry about inflated type II error? I encourage the authors to consider a correction for multiple testing and/or review if the few significant differences noted are all that meaningful given the number of tests and differences in terms of effect size.

Discussion

I appreciate the authors’ inclusion of thoughts on how trajectory groups can be difficult to replicate, especially under different populations of study, i.e., Netherlands vs. other European countries (also with different risk measures, urban vs. rural, etc.). However, there should be mention of how they may not generalize to other populations, such as adolescents in the US or Canada, etc. (as the literature reviewed included several US samples as I understand it).

There is a lot of mention on results of alcohol use but not on results of smoking, cannabis, and other risk behaviors. Thus, the discussion is not balanced with the results presented.

If alcohol use is not the best risk measure in relation to later outcomes, what is? Every other risk behavior tested? Is one perhaps especially more relevant given effect size? E.g. for Cannabis (table 6) - any cannabis use vs. low? Is heavy smoking vs. not show larger effects or smaller compared to cannabis? Externalizing behavior trajectories seem to have the largest effect in relation to no job.

6. PLOS authors have the option to publish the peer review history of their article (what does this mean?). If published, this will include your full peer review and any attached files.

Reviewer #1: Yes: Shuying Sha

Reviewer #2: No

---

## [Author Response · Author response to Decision Letter 0]

8 Oct 2019

Reviewer 1.

This paper has examined 1) the construct of risk behavior at adolescent period 2) the development trajectory of risk behaviors, and 3) the association between development trajectory and outcomes at early adulthood. The purpose, data collection (measurement) are well specified. The statistical methods overall are appropriate. The study did not find there is a constant construct underlying the measured risk behaviors across age through CFA. For each risk behavior, the study revealed different developmental trajectory through latent growth modeling. In addition, through logistic regressions, the study found statistically significant relationship between interested outcomes and trajectories of risk behaviors but not the other. 

The statistical methods applied are sufficient for the research purposes and the data properties, and the discussion of the missing data and its potential impact on investigated research questions is reasonable. Overall, the writing of the paper is well organized and clearly, though, there are a few places that needs clarification and improvement before publication. 

Method and Result presentation

1. First, the authors indicated that CFA was performed to test the construct of risk behavior. However, it is not clear what specific model is applied to the data of multiple waves. It is assumed that first a configural invariance of single-factor model is specified to see if there is an underlying latent construct for smoking, cannabis use, and alcohol use, etc. If this model fit is not sufficient, it is not necessary to further test metric invariance or scalar invariance. It would be helpful if a path diagram is presented to show the structure of the model. Similarly, it would be helpful to present a summary table which display the factor loadings and test of significance. 

We thank the reviewer for these suggestions. We indeed performed a confirmatory factor analyses for each wave separately, including all risky behavior. Model fit for all CFA’s were acceptable but factor loadings a bit low, particularly in late adolescence and young adulthood (ranging from .635 to .328). A next step in our approach was to test for MI. An overview of the specific results for each step of MI are presented in the supplementary material; the overall conclusion was that model fit was poor and factor loadings, particularly for alcohol use were low (< .20) for any form of MI. We have added a diagram to present our model and steps taken to be more clear about our approach. In addition, we have added a table displaying the factor loadings and model fit measures and added them to our manuscript as supplementary material. 

2. Second, the labels for Tables need to have more details. For example, it is unclear what the numbers on Tables 1-4 represent, mean (SD). Also, although the authors have described in measurement section, it would be helpful if the table can standard alone with the necessary information. For Tables 5-8, please label reference group for the outcome variable, and OR (CI) on the head of the table wherever it is needed. 

We fully agree with reviewer that relevant references are missing in the tables. We now have added this information such as referring to age in relation to the risk behavior in question (cf. alcohol 14 years), and references to mean and SD. See for instance page 16-18.

3. The author discussed the diverse definition of risk behavior in “Conceptualization of the risk behavior”. However, which conceptualization is adopted in this study is unclear. Is there other literature examining the structure of risk behavior construct using CFA? 

We have added in more detail the definition of risk behavior that we adopted for our current study. We have cited some other studies that have investigated risk behavior in a similar approach, see for instance page 5. 

“In this study we conceptualize risk behavior as behaviors that can be perceived as reckless and can have negative consequences for adolescent health (6-8).”

4. Second, “Risk” used in epidemiology is not the same as “risky decision making” are different. Only when a behavior/factor has potential negative impact on health outcome(s), that behavior is considered a risk behavior/factor. For example, “being sedentary” is a risk factor of obesity, but “being sedentary” is not “risk-taking”. 

We have added some sentences to be more clear that risk and risky decision making are not similar and that different disciplines use different definitions and operationalizations of risk behavior. We fully agree with the reviewer that the two are distinct. In fact, that is a crucial aspect of our reasoning. The cognitive process underlying risk behavior and actually engaging in risky behavior (i.e. behavior that has a negative impact on health) are different, though some studies assume that decision making processes are a good proxy measure for risk taking behavior. This study questions that assumption. We have added the following paragraph to be clearer about this (p 5):

“This overview illustrates that the conceptualization of risky behavior in adolescence depends on the field of interest. This variation in conceptualization might relate to the different perspectives about the development of risk behavior and its consequences for adolescent health (16,36). In this study we conceptualize risk behavior as behaviors that can be perceived as reckless and can have negative consequences for adolescent health.”

5. Third, if a behavior is not considered risk in some age or group, the association between this behavior and later outcomes might have different interpretation. 

Risk behavior as known in the developmental field of research entails any kind of behavior that possibly has a negative outcome for the individual, not only for health but also for academic performance. Alcohol use in certain age groups may be perceived as relatively normal and accepted. In our study we did not observe clear relationships between alcohol use and negative outcomes in young adulthood. We have highlighted this reasoning a bit more by including the following sentences to our discussion section (p 23).

“Alcohol use trajectories did not differentiate between adolescents developing successfully into young adults and adolescents who experienced difficulties in transitioning into adult roles. Thus drinking alcohol apparently does not present a risk for the pertinent outcomes. This may be due to the fact that drinking alcohol has been quite normative for adolescents in the Netherlands, in particular in the TRAILS-cohort (54). 

6. The authors need to provide more information about study motivation and implication. For example, what is the motivation for studying development trajectory of risk behaviors. How will the knowledge of different development trajectories help with policy making and intervention. 

We have added the following paragraph to our discussion section to share our ideas about the implication these finding have for policy making and intervention (p 26). 

“These result suggest that focusing on alcohol use in adolescence as possible marker for negative outcomes in young adulthood will not be the best approach to identify adolescents at risk for later problems in young adulthood. By no means we want to imply that the chosen outcomes are exhaustive in predicting positive outcomes in young adulthood, though we believe work and education are important markers for successful transition into young adulthood (20,21,22). For policy and intervention purposes, it may be more efficient to focus on other risky behaviors, such as cannabis use or externalizing problems. More particular, is may be wise to focus on the heavy, persistent trajectories of risky behaviors to identify the adolescents most at risk for being unsuccessful in their transition into young adulthood.”

7. Since age is an important factor in developmental trajectory, the author need to be more specific in review literature. For example, line 138-139, binge drinking at “what age” predicted lower school drop-out? It is also important to differentiate alcohol use and binge drinking, as some alcohol use is acceptable while binge drinking is not.

We have carefully read the manuscript and revised where necessary. 

See for instance line 138-139 and lines 140-147.

“Negative impact on educational attainment was found for smoking and drug use, whereas binge drinking predicted lower school drop-out among high school and college students (18-25 years). In line with the latter study, educational attainment in young adulthood (around 25 years) was more weakly associated with the frequency of alcohol use before the age of 17 than of cannabis use in three different Australian cohort studies (41)”.  

Review Comments to the Author

Reviewer 2.

The purpose of this study was to identify unique developmental trajectories of risk behaviors and associations with outcomes using a large, longitudinal study of Dutch adolescents (TRAILS). Peak risk behavior occurred in late adolescence (>19 years), thus this is where the largest differences in risk behavior trajectories was demonstrated. The authors conclude this likely influences the successful transition into higher educational attainment and gainful employment. The authors also note that alcohol did not covary with other risk behaviors (cannabis, smoking, externalizing) and present this as a key finding. There are many strengths of the study design, measures, and analyses. I have a few concerns I outline below that I believe if addressed, would strengthen the manuscript substantially.

1. The intro lays out the rationale well but ignores research that has demonstrated the clustering of risk and externalizing behaviors, including different kinds of substance use, is stronger in adolescence and dissipates over time (people specialize in one drug, etc.). See Vrieze et al (Am J Psychiatry 2012; 169:1073–1081) for a discussion on this. I encourage the authors to consider integration of this perspective into your rationale and findings.

We thank the reviewer for this interesting suggestion. We have briefly discussed some other studies that highlight the importance of age differences in the introduction (see for Mcgee and colleagues, 1992 and Defoe and colleagues, 2015 page 3 line, 57-58), with a similar approach as Vrieze et al. (2012). We have now added the Vrieze et al. (2012) paper to our discussion section as this study is important for the discussion about clustering of risk behavior.

See for instance page 3 were we already discuss these possible age differences

”Some researchers have raised this issue of age-dependent involvement in risk behaviors (12,14,16), and revealed that the underlying construct indeed varied with age (9).”

And page 22 which we have added:

“In a similar study (56) it was found that symptoms of nicotine, alcohol and cannabis dependence and abuse clearly clustered together in adolescence (14-17 years), but not so much in young adulthood (22-29 years).”

2. Method

The attrition analyses would be more helpful if the authors could comment on the nature of the effect size rather than focusing on p-values alone. How big of an effect might this have on generalizing results to the larger population?

We mistakenly copied the wrong column of the t-test, resulting in not meaningful t-test. We have changed this and added in addition Cohen’s d effect sizes. 

3. Were the risk outcomes (e.g., cannabis) log-transformed to better approximate normality assumptions? Cannabis in particular seems quite skewed.

The reviewer is right that cannabis and smoking were skewed. We used a negative binominal model to account for this. See page 14 for the explanation and reasoning.

“Because smoking as well as cannabis use included many zero counts and overdispersed data, we used a negative binominal model for these two risk behaviors (53).”

4. Was there any variability in race/ethnicity of the sample? This might be included as a covariate/confounder, if so.

A limitation of this study is that the sample population is not ethnically diverse (less than 10% of the parents had a minority background). We therefore did not include ethnicity as covariate but we do acknowledge that this may be a limitation of the study and added the following paragraph on page 25. 

“Third, future research could include a more ethnically diverse population (in our sample only 10% of the parents had a minority background) to investigate whether results are similar for other ethnic groups. Research shows that alcohol consumption for instance is less common among young adolescents with a minority background (59), possibly because of religious considerations.” 

5. Results

The authors should present more details about their CFA model where alcohol was dropped (what were the factor loadings for the other risk behaviors? What were the loadings for males vs. females?). Why not evaluate separate CFA models at the different ages of assessment to compare how these models fit at unique developmental periods? In reference to the Vrieze et al. article above, you may find good fit at some point in adolescence but not later in young adulthood.

We have added a table to the supplementary material including the factor loadings and a path diagram to illustrate each step in our approach. We indeed performed separate CFA models for each wave and our intention was to model trajectories over the course of adolescence. A first step in this approach was to identify one single factor for each wave. However, a CFA model with all behaviors together did not provide good factor loadings or model fit (see supplementary material table 1). We therefore decided to model each risk behavior separately, as there was no statistical justification to continue with a latent factor including all risk behaviors together. We have interpreted this finding in a similar way as Vrieze et al (2012) did, (see also comment 1) but our explanation is a bit different as Vrieze and colleagues focused on dependence and abuse symptoms which may differ from simple use of substances and externalizing behavior as in our study. 

6. For the group names for risk behavior, the ones labelled “moderate” seem to me would be better reflected with the name “moderate-increasing” (for alcohol, smoking) and the group labeled “moderate” for cannabis appears to just be “low” whereas “stable low” is essentially “never use.”

We have changed the labeling for smoking and alcohol from “moderate” to “moderate increasing” and for cannabis for “stable low”to “never use” and for “moderate” to “low”. 

7. The figures demonstrating the trajectory groups are extremely hard to read (fuzzy).

We used a different program to convert jpeg. files to an acceptable format for PlosOne, which has improved the quality considerably. 

8. Given the large number of tests (comparing each trajectory for each risk behavior group to one another in relation to the three education and employment outcomes), do the authors worry about inflated type II error? I encourage the authors to consider a correction for multiple testing and/or review if the few significant differences noted are all that meaningful given the number of tests and differences in terms of effect size.

We re-analyzed all logistic regressions while correcting for multiple testing (Bonferroni method). The results slightly changed for alcohol use (when controlling for covariates one significant effect was absent), but overall findings remained similar. 

9. Discussion

I appreciate the authors’ inclusion of thoughts on how trajectory groups can be difficult to replicate, especially under different populations of study, i.e., Netherlands vs. other European countries (also with different risk measures, urban vs. rural, etc.). However, there should be mention of how they may not generalize to other populations, such as adolescents in the US or Canada, etc. (as the literature reviewed included several US samples as I understand it).

We thank the reviewer for this suggestion. We have added the following sentence to our limitation section. See page 25. 

“Generalizability problems may also arise for adolescents in other countries as drinking cultures differ among countries and legal policies can have an impact on legalization of drinking at a certain age (54,55)” 

10. There is a lot of mention on results of alcohol use but not on results of smoking, cannabis, and other risk behaviors. Thus, the discussion is not balanced with the results presented.

We now have added some additional discussion on cannabis, smoking and externalizing behavior. See for instance page 24.

“For cannabis, smoking and externalizing trajectories, heavy engagement was associated with an increased likelihood for lower education and unemployment. These results remained significant after controlling for confounding variables such as parental education and being raised in a single parent household. This suggests that heavy cannabis use, externalizing behavior and smoking are possible indicators for less successful adult role transitioning in young adulthood. This negative impact could be a result of more direct effects of substance use and externalizing behavior on, for instance, educational attainment, truancy and school drop-out (22) or trough processes of affiliation with substance using peers (41). It would be interesting for future research to examine possible mediating factors that explain why particular heavy smoking, externalizing behavior and cannabis use have a negative impact on outcomes in young adulthood. Other markers of adulthood such as marriage, children and financial situation (23), could be additionally included, to investigate whether the negative impact of heavy cannabis, smoking and externalizing behavior affects other aspects of adulthood as well.”

11. If alcohol use is not the best risk measure in relation to later outcomes, what is? Every other risk behavior tested? Is one perhaps especially more relevant given effect size? E.g. for Cannabis (table 6) - any cannabis use vs. low? Is heavy smoking vs. not show larger effects or smaller compared to cannabis? Externalizing behavior trajectories seem to have the largest effect in relation to no job.

We have added an additional paragraph about implications for policy and intervention. We suggest that alcohol use is not a good marker for possible difficulties with transitioning to adult role later in adolescence and young adulthood and advice to focus on externalizing behavior and cannabis instead. See for instance page 26.

 “These result suggest that focusing on alcohol use in adolescence as possible marker for negative outcomes in young adulthood will not be the best approach to identify adolescents at risk for later problems in young adulthood. By no means we want to imply that the chosen outcomes are exhaustive in predicting positive outcomes in young adulthood, though we believe work and education are important markers for successful transition into young adulthood (20,21,22). For policy and intervention purposes, it may be more efficient to focus on other risky behaviors, such as cannabis use or externalizing problems. More particular, is may be wise to focus on the heavy, persistent trajectories of risky behaviors to identify the adolescents most at risk for being unsuccessful in their transition into young adulthood.”

---

## [Decision Letter · Decision Letter 1]

30 Oct 2019

Unique Developmental Trajectories of Risk Behaviors in Adolescence and Associated Outcomes in Young Adulthood

PONE-D-19-17958R1

Dear Dr. Peeters,

We are pleased to inform you that your manuscript has been judged scientifically suitable for publication and will be formally accepted for publication once it complies with all outstanding technical requirements.

With kind regards,

Geilson Lima Santana, M.D., Ph.D.

Academic Editor

PLOS ONE

Additional Editor Comments (optional):

Reviewers' comments:

Reviewer's Responses to Questions

**Comments to the Author**

1. If the authors have adequately addressed your comments raised in a previous round of review and you feel that this manuscript is now acceptable for publication, you may indicate that here to bypass the “Comments to the Author” section, enter your conflict of interest statement in the “Confidential to Editor” section, and submit your "Accept" recommendation.

Reviewer #1: All comments have been addressed

Reviewer #2: All comments have been addressed

2. Is the manuscript technically sound, and do the data support the conclusions?

Reviewer #1: Yes

Reviewer #2: Yes

3. Has the statistical analysis been performed appropriately and rigorously? 

Reviewer #1: Yes

Reviewer #2: Yes

4. Have the authors made all data underlying the findings in their manuscript fully available?

Reviewer #1: No

Reviewer #2: No

5. Is the manuscript presented in an intelligible fashion and written in standard English?

Reviewer #1: Yes

Reviewer #2: Yes

6. Review Comments to the Author

Reviewer #1: (No Response)

Reviewer #2: The authors have successfully responded to prior concerns and the manuscript has improved as a result. I recommend accepting this manuscript for publication.

7. PLOS authors have the option to publish the peer review history of their article (what does this mean?). If published, this will include your full peer review and any attached files.

Reviewer #1: No

Reviewer #2: No

---

## [Editor Report · Acceptance letter]

5 Nov 2019

PONE-D-19-17958R1 

Unique Developmental Trajectories of Risk Behaviors in Adolescence and Associated Outcomes in Young Adulthood 

Dear Dr. Peeters:

I am pleased to inform you that your manuscript has been deemed suitable for publication in PLOS ONE. Congratulations! Your manuscript is now with our production department. 

With kind regards,

on behalf of

Dr. Geilson Lima Santana 

Academic Editor

PLOS ONE